# Body Composition as a Predictor of the Survival in Anal Cancer

**DOI:** 10.3390/cancers14184521

**Published:** 2022-09-18

**Authors:** Ahmed Allam Mohamed, Kathrin Risse, Jennifer Stock, Alexander Heinzel, Felix M. Mottaghy, Philipp Bruners, Michael J. Eble

**Affiliations:** 1Department of Radiation Oncology, RWTH Aachen University Hospital, 52074 Aachen, Germany; 2Center for Integrated Oncology Aachen, Bonn, Cologne and Duesseldorf (CIO ABCD), 52074 Aachen, Germany; 3Department of Nuclear Medicine, RWTH Aachen University Hospital, 52074 Aachen, Germany; 4Department of Radiology and Nuclear Medicine, Maastricht University Medical Center (MUMC+), 6229 HX Maastricht, The Netherlands; 5Department of Diagnostic and Interventional Radiology, RWTH Aachen University Hospital, 52074 Aachen, Germany

**Keywords:** biomarker, anal malignancies, gastrointestinal tumors, obesity

## Abstract

**Simple Summary:**

Obesity and overweight are major health hazards in general and for cancer patients in specific. In this study, we used two parameters for measuring obesity, namely body mass index (BMI) and visceral-to-subcutaneous fat ratio (VSR), to evaluate the effect of obesity on the survival of patients with cancer of the anus. Our results show that obese patients live shortly compared to patients without obesity. Measures to counter obesity at the time of diagnosis could be necessary to improve the outcomes of treatment.

**Abstract:**

**Background and aim:** Sarcopenia and body composition parameters such as visceral and subcutaneous adipose tissue and visceral-to-subcutaneous adipose tissue ratio have been shown to be relevant biomarkers for prognosis in patients with different types of cancer. However, these findings have not been well studied in anal cancer to date. Therefore, the aim of this study was to evaluate the prognostic value of different body composition parameters in patients undergoing radiation therapy for the treatment of anal cancer with curative intent. **Material and Methods:** After approval by the institutional ethical committee, we retrospectively identified 81 patients in our local registry, who received radical intensity-modulated radiotherapy for the management of anal squamous cell cancer (ASCC). Clinical information, including body mass index (BMI), survival, and toxicities outcome, were retrieved from the local hospital registry. Based on the pre-therapeutic computer tomography (CT), we measured the total psoas muscle area, visceral adipose tissue area (VAT), subcutaneous adipose tissue area (SAT), and visceral-to-subcutaneous adipose tissue area ratio (VSR). In addition to the classical prognostic factors as T-stage, N-stage, gender, and treatment duration, we analyzed the impact of body composition on the prognosis in univariate and multivariate analyses. **Results:** Sarcopenia was not associated with increased mortality in anal cancer patients, whereas increased BMI (≥27 kg/m^2^) and VSR (≥0.45) were significantly associated with worsened overall survival and cancer-specific survival in both univariate and multivariate analyses. VSR—not BMI—was statistically higher in males. Sarcopenia and VSR ≥ 0.45 were associated with advanced T-stages. None of the body composition parameters resulted in a significant increase in treatment-related toxicities. **Conclusion**: BMI and visceral adiposity are independent prognostic factors for the survival of patients with anal cancer. Measurements to treat adiposity at the time of diagnosis may be needed to improve the survival outcomes for the affected patients.

## 1. Introduction

Recently, there has been a growing interest in the research field of ASCC as a result of the increasing incidence and mortality in addition to the recent advances in therapeutics, radiation and imaging techniques [1,2]. Intensity-modulated radiotherapy (IMRT) concurrent with fluoropyrimidines and mitomycin is by far the standard management of non-metastatic stages [3,4]. The one-size-fits-all strategy in the management of ASCC may result in overtreatment for early stages or undertreatment for advanced ones. Currently, different strategies are being tested for therapy de-escalation in the early stages and escalation for advanced tumors [5,6,7]. For this reason, it would be necessary to have prognostic tools to accurately define the high-risk patients who need the escalation of their therapy to be cured and the low-risk patients who could be spared the aggressive therapeutic approaches without compromising their survival outcomes. 

In the same perspective, the staging system in ASCC has been a matter of evolution and critique. The Nordic anal cancer group evaluated the prognostic strength of the 8th edition of the staging system of the Union for International Cancer Control (UICC) in 1151 patients retrospectively. The authors reported the overlapping of the different staging subgroups that reduces the reliability of the staging system to classify patients in different risk groups [8]. 

Additionally, finding reliable prognostic biomarkers would help define the high-risk patients who may profit from therapy escalation. Currently, there are few prognostic biomarkers that have been established in the management of ASCC. A recent systematic review by Theophanous et al. [9] addressed the topic, and they found that in addition to the tumor stage, nodal involvement, and gender, the following biomarkers could be of prognostic significance: baseline leukocytosis, neutrophilia, anemia, and human papilloma (HPV) virus load. 

On the other hand, the value of body composition parameters as biomarkers in cancer management has been evaluated in different studies [10,11]. These include sarcopenia, defined as progressive skeletal muscle wasting [12], which could reflect a state of cachexia and has emerged as a prognostic tool in multiple tumor sites [13]. Other parameters reflecting the body composition, including body mass index (BMI), visceral adipose tissue abdominal (VAT), subcutaneous abdominal adipose tissue (SAT), and visceral-to-subcutaneous fat ratio (VSR) have been also discussed as prognostic factors and predictors for the treatment-related toxicities in different solid tumors [14,15].

While the prevalence and prognostic effect of sarcopenia were examined in anal cancer in a few studies, other body composition parameters, namely obesity and fat distribution, were not tested. 

In this study, our aim was to evaluate the prognostic effect of BMI and other computer tomography (CT)-based biomarkers, including sarcopenia, obesity, and fat distribution in a cohort of patients treated with IMRT in curative settings. 

## 2. Material and Methods

This is a retrospective analysis that was approved by the local ethical committee of the Faculty of Medicine, RWTH Aachen University (EK 478-21). In this analysis, we included all patients with ASCC who were treated in the Department of Radiation Oncology at RWTH Aachen University Hospital, and received definitive radiochemotherapy (RCT) or radiotherapy (RT) with curative intent starting from January 2009 until December 2021. 

The eligibility criteria for the analysis were defined as follows: Histologically proven squamous cell cancer of anal canal;Stage I–III (union for international cancer control “UICC” 8th edition);The treatment was delivered as intensity-modulated radiotherapy (IMRT).

The exclusion criteria were defined as follows: Non-squamous cell cancer histology;Stage IV disease (UICC 8th edition);Radiation was applied as conventional or conformal three-dimensional radiotherapy (3DRT);Radiation was applied as local palliative treatment.

The clinical data examined included age, sex, stage (UICC 8th edition), lymph node status, weight, height, BMI, body surface area (BSA), locoregional relapse-free survival (LRFS), which was defined as the time span from the end of the therapy until the occurrence of any disease progression or recurrence in the pelvis or being censored, disease-free survival (DFS), which was defined as the time span from the end of the therapy until the occurrence of any type of relapse (locoregional or metastatic) or being censored, overall survival (OS), which was defined as the time span from the diagnosis until death from any cause and cancer-specific survival (CSS), which was defined as the time span from the diagnosis until death from anal cancer or treatment or being censored. Toxicities were graded based on the Common Terminology Criteria for Adverse Events (CTCAE) Criteria 5.0.

### 2.1. Image Analysis

Before treatment initiation, all patients underwent contrast-enhanced multislice planning computer tomography (P-CT) on a 16-slice CT scanner (from January 2009–July 2016: Somatom Emotion, Siemens, Erlangen, Germany, and from August 2016–December 2021: Brilliance CT Big Bore Oncology, Philips Medical Systems Inc., Cleveland, OH, USA) or planning positron emission tomography/computer tomography (PET/CT) on Gemini TF 16 positron emission tomography/computer tomography, Philips Medical Systems, Best, The Netherlands, using 120 kVp slice thickness of 3 mm (reconstructed pixel size 1.17 mm × 1.17 mm). For further analysis, P-CTs were exported in DICOM format to the 3D Slicer segmentation program [16]. All measurements were carried out on a single image at the intervertebral disc level between lumbar vertebrae L4-L5. For segmentation purposes, the following CT attenuation thresholds used were from 150 to −29 Hounsfield Unite (HU), from −50 to −150 HU, and from −30 to −190 HU for semi-automated segmentation of the total psoas muscle area (TPA), VAT, and SAT, respectively, using the masking tool in the program [17] (Appendix A).

The visceral-to-subcutaneous fat ratio (VSR) was calculated using the following formula: $$\frac{VAT}{SAT}$$

The total psoas area index (TPAI) was calculated using the following formula:TPAheight ^2

The sarcopenia was diagnosed when TPAI was below 7.5 and 5.2 cm^2^/m^2^ in males and females, respectively, as previously defined [18]. 

To minimize the measurement error, two investigators independently performed all measurements (AM, KR), and the mean value of both measurements was used for further analyses. 

### 2.2. Statistical Analysis

The Pearson correlation coefficient was used to measure the correlation between BMI, VAT, SAT, and VSR. An individual time-dependent receiver operating characteristic (ROC) curve analysis was used to detect the cutoff point of significance for continuous variables (BMI and VSR) based on OS. The chi-square test was used to compare the non-parametric variables, and the unpaired T-test was used to compare the mean of the two groups. The Kaplan–Meier survival analysis with log-rank test was used for univariate analysis, and the Cox regression analysis was applied for the multivariate analysis. The statistical analyses and graphs were made using R-software version 4.0.2 (R Foundation for Statistical Computing, Vienna, Austria).

## 3. Results 

### 3.1. Patients’ Characteristics

We identified 93 patients with histologically proven ASCC who were treated in our department between January 2009 and December 2021. Twelve patients were excluded from the analysis due to: 

Upstaging in stage IV through further diagnostics before the beginning of treatment in 10 patients;

Unavailability of both imaging and clinical information in two patients; 

Eighty-one patients were eligible for the analysis (48 females and 33 males) with a median follow-up of 28.4 months (range 5–124 months)—the baseline characteristics are shown in Table 1. 

The 5-year-LRFS, DFS, OS and CSS for the whole cohort of patients were 80.1% (CI: 70–91.1%), 77.2% (CI 67.2–88.6%), 82.8% (CI: 73.4–93.3%) and 82.8% (73.4–93.3%), respectively. The median radiation dose to the primary tumor was 56 Gy. Concurrent chemoradiation was applied in 74 patients (91.3%), while 7 patients (8.7%) only received radiation in curative settings (three patients with stage I and four patients rejected the chemotherapy due to other comorbidities).

The median BMI, SAT, VAT and VSR was 25 kg/m^2^ (range: 17.2−38 kg/m^2^), 323 cm^2^ (range: 821–36.6 cm^2^), 157 cm^2^ (range: 414.6–11.5 cm^2^), and 0.45 (range: 1.74–0.11), respectively; Sarcopenia was diagnosed in 14 patients (17.7%) (Table 1). The median deviation between the mean value and the measurements of the two observers for VSR was 0.031 (range: 0–0.28).

There was a strong correlation between BMI and both SAT and VAT (r = 0.76 and 0.6, *p*-value < 0.0001) and weak correlation between the SAT and VAT (r = 0.44, *p*-value < 0.0001). However, there was no correlation between the BMI and VSR (r = 0.023, *p*-value = 0.84) (Figure 1). Furthermore, there was a strong correlation between BSA and BMI (r = 0.77, *p* < 0.0001) and weak correlation between BSA and VSR (r = 0.21, *p* = 0.07) (Figure 1).

The median BMI was not statistically different in males and females, 25.69 and 24.75 kg/m^2^ (*p*-value = 0.29), respectively. Furthermore, the median of VSR in males was significantly higher compared to the median in females (0.64 vs. 0.33, respectively, *p*-value < 0.00001) (Figure 2). 

### 3.2. The Univariate Analysis

The presence of sarcopenia did not translate into a worsening of the OS, CSS, DFS, or LRFS (*p*-value 0.84, 0.84, 0.92, and 0.81, respectively) (Table 2, Figure 3). However, Sarcopenia was frequently associated with T3-4 stages (11 patients, *p*-value: 0.002), while there was no significant association with the nodal status (*p*-value: 0.172, Table 3). 

Furthermore, the time-dependent ROC curve analysis was used to identify a cut-off point with the best combination of sensitivity and specificity for OS using BMI. The area under the curve (AUC) was 0.73 with a cutoff point of 27 kg/m^2^, showing a sensitivity and specificity of 66.7% and 73.5%, respectively. In the univariate analysis using the log-rank test, patients with BMI < 27 kg/m^2^ had statistically significant better OS, CSS, DFS, and LRFS (*p*-value 0.0033, 0.0013, 0.003, and 0.0024, respectively) (Table 2, Figure 3). A BMI of ≥27 kg/m^2^ was not associated with higher T3-4 or N1 stages (*p*-value: 0.58 and 59, respectively) (Table 3).

Similarly, a cutoff point of 0.45 was identified for VSR, with an AUC of 0.71 and sensitivity and specificity of 83.3% and 53.6%, respectively. A VSR below 0.45 was associated with favorable OS and CSS (*p*-value 0.0041 and 0.01, respectively) in the univariate analysis. However, this did not reach statical significance for LRFS and DFS (*p*-value 0.11 and 0.23, respectively) (Table 2, Figure 3). Additionally, a VSR of ≥0.45 was associated with higher T-stages *(p*-value: <0.001), while there was no statistically significant association with the nodal status (*p*-value: 0.46) (Table 3). 

Other parameters that were associated with improved CSS in univariate analyses were female sex and treatment duration ≤44 days (*p*-value: 0.008 and 0.047, respectively) (Table 2). 

Larger tumor size, lymph node involvement, or advanced stage was not associated with the statistically significant worsening of any of the survival parameters in the current analysis (Table 2) (Figure 3).

### 3.3. Multivariate Analysis

A Cox regression analysis for a model includes the following variables: the presence or absence of Sarcopenia, BMI (≥27 vs. <27 kg/m^2^), VSR (≥0.45 vs. <0.45), and early (T1,2+N0) vs. the advanced (T3,4 or N1) stage was evaluated. 

BMI < 27 was associated with improved OS, CSS, DFS, and LRFS (Table 4). Furthermore, VSR < 0.45 only showed a statistically significant improvement in OS and CSS. Sarcopenia and advanced staged was not associated with the worsening of any of the survival parameters (Figure 4) (Table 4). 

### 3.4. Toxicity Analysis

Sarcopenia was neither associated with an increase in grade 3 hematological or gastrointestinal toxicities (*p*-value = 0.67 and 0.88, respectively) nor resulted in the frequent interruption of the treatment as well (*p*-value = 0.59).

Similarly for BMI ≥ 27 kg/m^2^ and VSR ≥0.45, there was no significant increase in hematological or gastrointestinal toxicities (*p*-value = 0.87 and 0.16 and 0.22 and 0.53, respectively). Furthermore, neither BMI ≥27 kg/m^2^ nor VSR ≥0.45 were associated significantly with treatment interruption (*p*-value = 0.35 and 0.082, respectively).

## 4. Discussion

Obesity and abdominal fat distribution have been evaluated in previous studies as possible prognostic biomarkers in various tumors [19,20]. However, to the best of our knowledge, they have not been evaluated regarding their impact on the survival of anal cancer. 

In addition to cardiovascular diseases and diabetes mellitus, being overweight or obese has been linked to a higher incidence of cancer [21]. A recent meta-analysis of 203 studies found that obesity was associated with higher cancer mortality [22]. BMI has been the simplest tool to define overweight and obesity. However, there has been some evidence that BMI may not be the most accurate tool for this purpose, with some discrepancies between obesity and BMI [23]. 

Moreover, there has been a growing body of evidence that the distribution of adipose tissue could be of significance to cancer initiation and progression, with visceral abdominal adipose tissue being acknowledged as the most harmful type of adipose tissue [24]. The possible reasons for the pro-oncogenic character of visceral adipose tissue could be related to inducing insulin resistance which results in hyperinsulinemia and possibly metabolic syndrome [25]. Nevertheless, there is evidence that adiposity induces a state of hypoxia, which includes the hypoxia of adipose tissue itself [26]. Furthermore, the hypoxia of the visceral adipocyte but not the peripheral adipocyte would boost the production of the hypoxia factor 1α (HIF1α) and inflammation, a state that would stimulate tumor progression [27]. VSR is a ratio that describes the distribution of abdominal adipose tissue and has been described as a prognostic tool in other cancers such as colorectal and uterine cancer [10,28]. 

On the level of the tumor microenvironment, obesity impairs CD8^+^ T cell, through increased fat uptake in tumor cells and a reduction in the tumor-infiltration of CD8^+^ T cell and its function [29].

In the current study, we included patients from our register who were only treated with IMRT due to the possible influence of the treating modality on survival and the availability of imaging data [30]. Our results show a correlation between BMI with both visceral and subcutaneous adipose tissue but did not show any correlation between BMI and VSR; this may indicate that VSR could provide a different prognostic value than BMI. 

Additionally, there was a correlation between BSA and BMI and to a lesser extent, between BSA and VSR. This correlation should be acknowledged in light of the controversies around the dose calculation of the chemotherapy based on BSA, which is associated with great interpatient and intrapatient drug-plasma level variations [31]. In the same context, obesity can alter the metabolism of chemotherapeutics. In a recent in vivo study, Vanderveen et al., reported that obesity alters the metabolism of 5-fluorouracil (5FU), significantly reducing the survival in obese mice receiving the drug, reducing dihydropyridine dehydrogenase (DPD) activity, the enzyme responsible for 5FU metabolism and increase the inflammatory gene signature in peripheral tissues [32].

Furthermore, the impact of sarcopenia on treatment outcomes has been frequently studied in various cancers, with results suggesting inferior survival or increased treatment-related morbidities for patients with sarcopenia [11]. Recently, the effect of sarcopenia on survival has been replicated in a handful of studies in anal cancer with conflicting results. Some studies have linked sarcopenia with an inferior survival [33,34]. On the other hand, Martin et al. could not confirm a prognostic role for sarcopenia in anal cancer. However, they described the association of increased hematological toxicities with sarcopenia in patients who underwent the RCT [35].

In the current study, sarcopenia was diagnosed in 17.7% of the patients. However, this did not translate into inferior survival outcomes in the univariate analysis. On the other hand, a cutoff for BMI and VSR of 27 kg/m^2^ and 0.45, respectively, was applied in this study based on an individual time-dependent ROC-curve analysis for OS. Both BMI ≥ 27 kg/m^2^ and VSR ≥ 0.45 were associated with poor OS and CSS in the univariate analysis. Both Sarcopenia and higher VSR were associated with more advanced T-stages but not N-stages. Other factors associated with deteriorating survival were the male gender and treatment duration > 44 days. 

In the multivariant analysis, BMI < 27 and VSR < 0.45, showed statistically significant improvement in OS and CSS. It is surprising that none of the classical prognostic factors, such as primary tumor size, nodal involvement, or tumor stage, were statistically significant in the univariate or multivariate analysis. This could be attributed to the relatively lower number of patients included in the analysis. However, other analyses for the validation of the 8th edition of UICC TNM staging with a larger cohort concluded similar overlaps for the survival of different subgroups based on this edition of the classification [8]. 

In light of these results, obesity and visceral adiposity are relevant risk factors that need to be addressed in the initial assessments of patients and in future risk stratification of the disease. 

Interestingly, the median of VSR in males was significantly higher than in females; however, this was different in the case of BMI, where there were no gender-related differences. This phenomenon may help explain the difference in OS and CSS between males and females in anal cancer that was reported in the survival analysis in this study and a previous analysis from our group with a larger collective of patients [30]. 

An association between increased hematological or gastrointestinal toxicities during the treatment and sarcopenia, BMI, or VSR could not be established in our study.

## 5. Study Limitations

There are some limitations of the current study that have to be openly addressed. The main limitation is the retrospective nature of the study, which could make the analysis prone to different types of bias which may impact the results. Furthermore, the incidence of HPV-associated ASCC has grown, accounting for almost 75% of the total incidences of ASCC [36]. It has been reported in frequent studies that HPV-associated ASCC tends to have a better prognosis in comparison to other ASCC [37]. Due to the current lack of therapeutic implications of HPV status, this has not been systematically evaluated in most patients in the current study before treatment initiation. 

## 6. Conclusions

The obesity and higher visceral adipose tissue content in patients with anal cancer resulted in inferior survival outcomes after treatment. In light of the results of this study, a careful baseline evaluation of patients, including the status of obesity and fat distribution analysis with active countermeasures in case of higher visceral fat content or general adiposity, may be necessary for the initial steps of anal cancer management to improve the survival. Further validation of the results is needed.

## Figures and Tables

**Figure 1 cancers-14-04521-f001:**
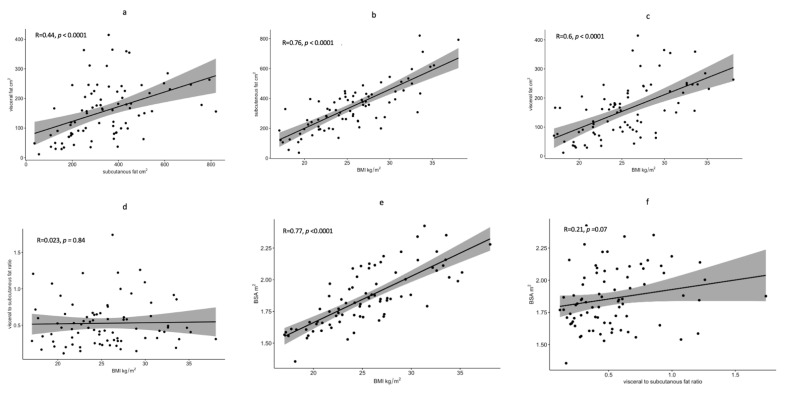
Scatter plot diagrams showing the Correlation between SAT and VAT (**a**); between BMI and SAT (**b**); between BMI and VAT (**c**); between BMI and VSR (**d**); between BSA and BMI (**e**); and BSA and VSR (**f**), black dots represent measurements from 81 patients, the solid line represents the regression line and grey shadow represents the confidence interval.

**Figure 2 cancers-14-04521-f002:**
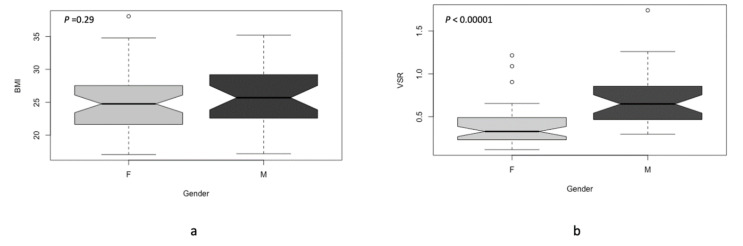
(**a**) Boxplot diagram showing BMI in both males “M” and females “F” and diagram (**b**) showing VSR in males “M” and females “F”.

**Figure 3 cancers-14-04521-f003:**
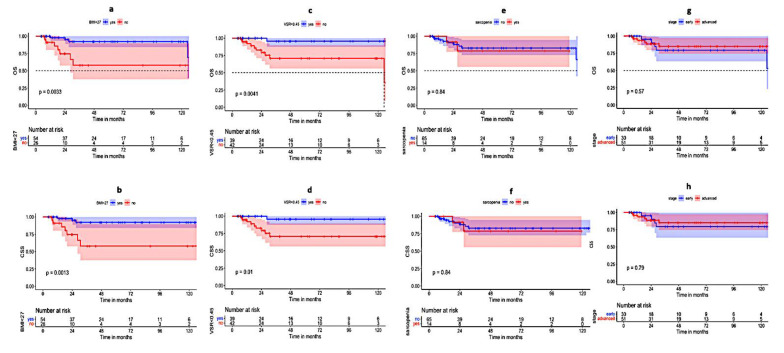
The Kaplan–Meier survival curves showing the difference in OS and CSS for patients with ASCC based on BMI (≥27 vs. <27 kg/m^2^) (**a**,**b**); based on VSR (≥0.45 vs. <0.45) (**c**,**d**); based on Sarcopenia (yes vs. no) (**e**,**f**); and tumor stage (advanced vs. early) (**g**,**h**), in red and blue, respectively.

**Figure 4 cancers-14-04521-f004:**
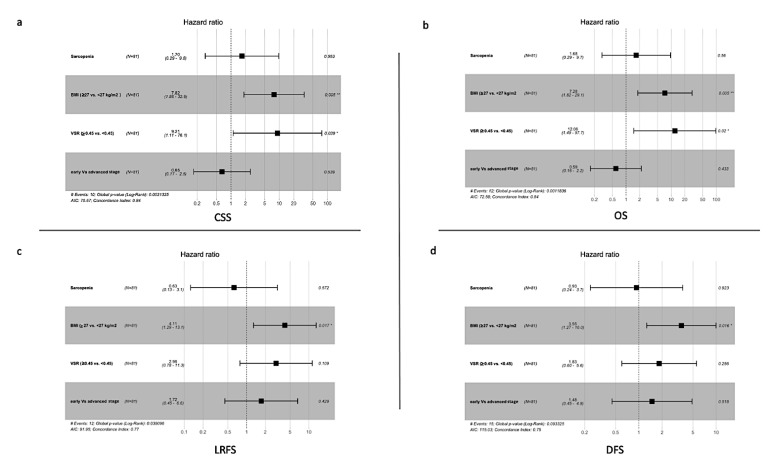
COX hazard regression analysis of Sarcopenia, BMI, VSR, and tumor stage for CSS (**a**), OS (**b**), LRFS (**c**), and DFS (**d**), showing the hazard ratio for each parameter, *: *p* value < 0.05, **: *p* value ≤ 0.005.

**Table 1 cancers-14-04521-t001:** Baseline characteristics—BMI: body mass index; BSA: body surface area; VSR: visceral-to-subcutaneous fat ratio; VAT: visceral adipose tissue abdominal; SAT: subcutaneous abdominal adipose tissue; UICC: Union for International Cancer Control.

Characteristics	
Sex	
Female	48 (59.3%)
Male	33 (40.7%)
Age: median/range	58 (35–86)
BMI: median/range	25 kg/m^2^ (17.2–38 kg/m^2^)
BSA: median/range	1.83 m^2^ (1.35–2.4 m^2^)
SAT: median/ range	323 cm^2^ (36.6–821 cm^2^)
VAT: median/range	157 cm^2^ (11.5–414.6 cm^2^)
VSR: median/range	0.45 (0.11–1.74)
Sarcopenia:	
Yes	14
No	65
undetermined	2
cT-Status	
1	15
2	34
3	22
4	9
Undetermined	1
cN-status	
0	40
1	41
Stage (UICC TNM 8th, 2017)	
I	10
IIA	21
IIB	7
IIIA	20
IIIB	3
IIIC	20
P-16 status	
Positive	8
Negative	3
Undetermined	70
Concurrent chemotherapy	
Yes	74
No	7

**Table 2 cancers-14-04521-t002:** BMI: body mass index; VSR: visceral-to-subcutaneous fat ratio univariate analysis using the log-rank test. * *p*-value < 0.05: significant.

Parameter	LRFS	DFS	OS	CSS
**Sex (Female vs. Male)**	0.079	0.129	0.016 *	0.008 *
**T1 + 2 vs. T3 + 4**	0.089	0.124	0.403	0.403
**Nodal involvement (N0 vs. N1)**	0.26	0.243	0.66	0.885
**early (T1-2,N0) vs. advanced (T3,4 orN1) stage**	0.43	0.47	0.57	0.79
**Sarcopenia (No vs. Yes)**	0.81	0.92	0.84	0.84
**BMI (<27 vs. ≥27 kg/m^2^)**	0.0024 *	0.003 *	0.0013 *	0.0033 *
**VSR (<0.45 vs. ≥0.45)**	0.11	0.23	0.0041 *	0.001 *
**Treatment duration (≤44 days vs. >44 days)**	0.169	0.142	0.132	0.047 *

**Table 3 cancers-14-04521-t003:** BMI: body mass index; VSR: visceral-to-subcutaneous fat ratio; ^$^ chi square test, * *p*-value < 0.05: significant.

Parameter No. of Patients	T1-2	T3-4	*p*-Value ^$^	N0	N1	*p*-Value ^$^
**Sarcopenia** (14 patients)	3	11	0.002	5	9	0.172
**BMI ≥ 27 kg/m^2^** (26 patients)	16	10	0.58	13	13	0.59
**VSR ≥ 0.45)** (42 patients)	18	24	<0.001 *	20	22	0.46

**Table 4 cancers-14-04521-t004:** BMI: body mass index; VSR: visceral-to-subcutaneous fat ratio, Cox regression analysis, * *p*-value < 0.05: significant.

Parameter	LRFS	DFS	OS	CSS
**Early (T1,2 N0) vs. advanced (T3,4 or N1) stage**	0.429	0.518	0.433	0.539
**Sarcopenia (No vs. Yes)**	0.572	0.923	0.56	0.553
**VSR (<0.45 vs. ≥0.45)**	0.109	0.286	0.02 *	0.039 *
**BMI (<27 vs. ≥27 kg/m^2^)**	0.017 *	0.016 *	0.005 *	0.005 *

## Data Availability

The data presented in this study are available upon request from the corresponding author.

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
