# Peer review of "Body Composition as a Predictor of the Survival in Anal Cancer"

_cancers, 2022, doi:10.3390/cancers14184521_

Round 1
Reviewer 1 Report
In this study, Ahmed Allam Mohamed and collaborators evaluated the prognostic value of different body composition parameters in anal cancer patients undergoing radiation therapy. Although this study is based only on retrospective analysis and further validation of the results is required, the authors found that BMI and visceral adiposity are independent prognostic factors for the survival of the affected patients. In contrast, sarcopenia was not associated with increased mortality in the indicated cancer patients. They concluded that measurements to treat adiposity at the time of the diagnosis may be needed to improve the survival outcomes of patients with anal cancer.
The manuscript is of interest and well written.
Comments/Questions:
Since tumor microenvironment play a crucial role on tumor progression, is there any correlation between BMI, visceral adiposity and immune cell infiltration in anal cancer patients survival?
Author Response
Thank you for the comment. We modified the discussion to include the literature that links obesity and immune cell infiltration (line 439-441). However, we didn’t find specific data for anal cancer. We will be interested in the future to explore this in our patients' cohort, however, this would take time to be achieved.

Reviewer 2 Report
The manuscript "Body composition as a predictor of survival in anal cancer" adresses an important and currentyl underresearched topic in anal cancer. The manuscript is well written but there are some minor complaints/issues that should be adressed in order to make the paper suitable for publication.
1. Table 1: The numbers for the range of Age, SAT, VAT and vsr seem to be switched.
2. Line 129-131: Could the authors provide some measurements of how far the investigators deviated from each other? This would be important to show how robust the used methodology was for this study.
3. Line 168-171: I would not deem r=0.44 a strong correlation. Either remove the word completely or add something like "weak" for the correlation between SAT and VAT.
4. It is kind of surprising that there is no impact of established parameters like T and N stage on outcome. This merits a discussion of possible reasons for this. An important point to discuss is how to interpret other significant prognostic parameters if all the standard parameters are negative.
5. Multivariate Analysis: Generally i have some concerns creating a model with variables that are so heavily correlated. This can create problems with multicollinearity. I would suggest to review this with a statistician - maybe you will need to remove some variables.
6. How many events do you have for OS, CSS, DSS and LRFS? There is a rule of ten for multivariate cox regression analysis that suggests that you need 10 events per added variable. While there is evidence that this rule can be relaxed ( DOI: 10.1093/aje/kwk052 ) i doubt that it is possible to the extent analyzed in the manuscript. I would first of all suggest to remove SAT and VAT as these variables are already incorporated into VSR. I would also suggest to show the hazard ratios besides the p values - maybe as a supplementary table.
7. The authors correctly discuss possible implications of different adipose tissues on inflammation and so on. Could there also be a role of differences regarding chemotherapy metabolism? I think it is clear that the long time standard of dosing per body surface area is not the most suitable for many patients. Could the authors show how BSA correlates with their adipose tissue measurements? Maybe this could also lead to a possible explanation of the differences in oncological outcome.
Author Response
The manuscript "Body composition as a predictor of survival in anal cancer" adresses an important and currentyl underresearched topic in anal cancer. The manuscript is well written but there are some minor complaints/issues that should be adressed in order to make the paper suitable for publication.
1. Table 1: The numbers for the range of Age, SAT, VAT and vsr seem to be switched.
Answer: Thank you, we corrected that.
Line 129-131: Could the authors provide some measurements of how far the investigators deviated from each other? This would be important to show how robust the used methodology was for this study.
Answer: We included the deviation of the 2 measurements from the mean in VSR in the result section (line 259-261)
3. Line 168-171: I would not deem r=0.44 a strong correlation. Either remove the word completely or add something like "weak" for the correlation between SAT and VAT.
Answer: we corrected that.
It is kind of surprising that there is no impact of established parameters like T and N stage on outcome. This merits a discussion of possible reasons for this. An important point to discuss is how to interpret other significant prognostic parameters if all the standard parameters are negative.
Answer: we totally agree; this has confused us, too, we edited the discussion part to discuss this point and its possible implications (line 473 to 478)
Multivariate Analysis: Generally i have some concerns creating a model with variables that are so heavily correlated. This can create problems with multicollinearity. I would suggest to review this with a statistician - maybe you will need to remove some variables.
6. How many events do you have for OS, CSS, DSS and LRFS? There is a rule of ten for multivariate cox regression analysis that suggests that you need 10 events per added variable. While there is evidence that this rule can be relaxed ( DOI: 10.1093/aje/kwk052 ) i doubt that it is possible to the extent analyzed in the manuscript. I would first of all suggest to remove SAT and VAT as these variables are already incorporated into VSR. I would also suggest to show the hazard ratios besides the p values - maybe as a supplementary table.
Answer: we would like to thank the reviewer for explaining this problem in the analysis. This point was not clear to us during the analysis. We repeated the cox regression analysis with only 4 essential parameters (Sarcopenia , BMI, VSR and tumor stage “early and advanced”). We included a new figure (4) that includes the hazard ratio and the number of events for each survival parameter.
The authors correctly discuss possible implications of different adipose tissues on inflammation and so on. Could there also be a role of differences regarding chemotherapy metabolism? I think it is clear that the long time standard of dosing per body surface area is not the most suitable for many patients. Could the authors show how BSA correlates with their adipose tissue measurements? Maybe this could also lead to a possible explanation of the differences in oncological outcome.
Answer: Again, we would like to thank the reviewer for addressing this point. We did the correlation of BSA and VSR and BMI, and we also included this point in the discussion with some new references that could actually explain the defect metabolism of 5FU in obese patients with inferior survival as an implication (line 447-455)
